# The Psychological Well-Being of Older Chinese Immigrants in Canada amidst COVID-19: The Role of Loneliness, Social Support, and Acculturation

**DOI:** 10.3390/ijerph19148612

**Published:** 2022-07-15

**Authors:** Chang Su, Lixia Yang, Linying Dong, Weiguo Zhang

**Affiliations:** 1Department of Psychology, Toronto Metropolitan University, Toronto, ON M5B 2K3, Canada; 2Ted Rogers School of Information Technology Management, Toronto Metropolitan University, Toronto, ON M5B 2K3, Canada; ldong@ryerson.ca; 3Department of Sociology, University of Toronto, Toronto, ON M5S 2J4, Canada; weiguo.zhang@utoronto.ca

**Keywords:** social support, loneliness, life satisfaction, psychological well-being, older Chinese immigrants, COVID-19

## Abstract

This study examined the effects of loneliness, social support, and acculturation on psychological well-being, as indexed by general emotional well-being and life satisfaction, of older Chinese adults living in Canada during the COVID-19 pandemic. A total of 168 older Chinese adults, recruited via WeChat and the internet, completed an online study through a facilitated Zoom or phone meeting, or through a website link, individually or in a group. The testing package included demographic information, The UCLA Loneliness Scale, The Multidimensional Perceived Social Support Scale, Vancouver Index of Acculturation, The Satisfaction with Life Scale, and The World Health Organization’s Five Well-Being Index. The results showed that the psychological well-being (both general emotional well-being and cognitively perceived life satisfaction) was positively predicted by perceived social support but negatively predicted by loneliness. Acculturation was not predictive of both outcomes, and it did not moderate the predictive relationships of social support or loneliness. The results shed light on the importance of community services that target enhancing social support and reducing loneliness in promoting psychological well-being of older Chinese immigrants in Canada amidst and post the pandemic.

## 1. Introduction

Since early 2020, COVID-19 has swept across the world and become a global public health crisis [1]. Unprecedented lifestyle restrictions and lockdown measures (e.g., social distancing, and quarantine) have been implemented globally to constrict the spread of the virus [2]. These measures, however, significantly reduced people’s social engagement and increased their loneliness, thus contributing to increased mental health issues [3,4]. There were age disparities in mental distress, perceived adversities, resilience, and each coping strategy over time by age group [5] regarding the detrimental effects of COVID-19. Compared with their young counterparts, older adults are the vulnerable population since their weakened/declined immune system makes them easily prone to infection, and they are perceived as being at the highest risk of COVID-19 [5,6]. Older adults, particularly those aged 80 or above and living in long-term care, are the most vulnerable to infection, severe illness, and death [7]. The rapid COVID-19 transmission, increased mortality rate, social distancing, and quarantine could exacerbate seniors’ mental health problems and disrupt their psychological well-being [8,9,10]. However, a recent study found that older adults showed more resilience in emotional regulation and problem-solving under COVID-19 than their young counterparts [5].

Psychological well-being can be described as general positive feelings and perceptions of life in emotional, social, and physical domains. It could also be identified as the perception of life satisfaction [11]. Life satisfaction, according to Diener [12], is a cognitive component of subjective psychological well-being. It represents an aspect of healthy and successful aging and directly influences the quality of life of older adults [13]. Many factors, such as a sense of dignity or control, independence, physical health, and social interaction, could affect the psychological well-being of older adults in Canada [14]. Social networks and healthy lifestyles play an important role in older adults’ life satisfaction and psychological well-being [15,16,17]. During COVID-19, older adults were at the highest risk to develop severe complications in Canada, and 33% of older adults reported their mental health was worse than before the pandemic [7]. As a result of reduced perceived social support, lack of community belonging, and barriers to using online technology or having internet access, older adults’ psychological well-being during the COVID-19 pandemic suffers [9]. Past studies showed that life satisfaction declined with aging, which might be due to ageism and other discriminations related to aging [18,19]. Many psychological factors, such as the sense of belonging, social support, identity and sense of community, loneliness, and discrimination contribute to life satisfaction [12,18,20]. A recent study showed that life satisfaction declined during the COVID-19 pandemic in Canada and was lowest among Asian immigrants and the highest among Canadian-born individuals [21]. To have a comprehensive understanding of psychological well-being, we assessed psychological well-being with two measures, one for the global emotional well-being component and the other for the cognitive component (i.e., life satisfaction) in the current study.

The outbreak of COVID-19 dramatically changed the way that people socialize. Social distancing and lockdown measures increased the number of older adults in social isolation and heightened feelings of loneliness, which are well-known risk factors that influence older adults’ physical health and psychological well-being [22,23]. Loneliness, a subjective feeling of distress derived from the discrepancy between the desired and the actual personal social network [24,25], negatively affects health, quality of life, well-being, and life satisfaction [26,27,28]. Multiple risk factors, such as lack of networks, widowhood, stressful life transition, and health problems, contribute to loneliness [29,30,31,32]. It becomes a major social or important health and mental health concern among older adults [33,34]. Loneliness and social disconnection contribute to anxiety and depression symptoms [12]. Quarantine and social distancing during COVID-19 further enhanced their feelings of loneliness. As a result, older adults feel disconnected or dissatisfied with the existing network of family and friends and restricted their social engagement [9].

A recent study demonstrated that social support can mediate social isolation and improve mental health status under COVID-19 [35]. Social support plays an important role in life satisfaction and mental health in later years [36,37], and it has a protective effect on mental health of immigrants in North America [38,39,40,41]. Social support is defined as verbal or non-verbal information, advice, tangible aid, or action offered by social intimates or inferred by their presence [42]. Social support could be formal or informal, including emotional, instrumental, and tangible [37]. Specifically, it has a great impact on the well-being of the Chinese elderly population, which advocates family and networking support in the Chinese culture [43]. During COVID-19, the quality of life of older adults was negatively impacted by reduced in-person social interactions and engagement. Older adults with health conditions, those living in private households, with a lower income, suffering from ageism and racism, or having mental health problems, had more of a need for social support than their healthier counterparts [7,44].

Immigrants may encounter numerous difficulties and adjustment challenges in the process of adapting to living in a new country [40,45]. The process of immigration and resettlement compromises acculturation and enculturation for individuals [46]. Acculturation refers to the endorsement of the host culture, whereas enculturation refers to the process of maintenance of the heritage culture [46,47]. Past research has demonstrated that higher acculturation to the host culture was related to better mental health in elderly Asians [47] and mitigated the effects of perceived stress on depressive symptoms of older Chinese adults in the United States [48]. Similarly, lower acculturation was related to lower quality of life among older Chinese immigrants [49].

Older Chinese immigrants represent one of the largest growing visible minority groups in Western countries, and most of them live in urban areas [50,51,52]. Chinese senior immigrants commonly report cultural, language, service, and health belief barriers that may negatively impact their health [53,54,55], such as acculturation problems, poor health, low social support or participation, low financial security, and disadvantaged socioeconomic status [56,57,58]. Many of them come to Canada to support their adult children on whom they primarily depend financially, physically, or emotionally [38,39,58]. They join recreational or social group activities to gain a sense of belonging [59], but this has become a challenge due to COVID-19 [12]. However, Chinese culture encourages family and close networking support which might buffer the detrimental effects of COVID-19. To our knowledge, little quantitative research has been conducted to systematically examine the effects of social factors (e.g., loneliness and social support) on the psychological well-being and life satisfaction of older Chinese adults in Canada under COVID-19.

Given the impact of COVID-19, the reported relationship between social support or loneliness and mental health, and the vulnerability of older Chinese immigrants [27,36], the current study aimed to (1) examine the effects of loneliness (independent variable, likely a risk factor) or social support (independent variable, likely a buffer) on psychological well-being, as indexed by the global emotional well-being and life satisfaction (dependent variables) of older Chinese immigrants in Canada during the pandemic; and (2) explore the moderating effects of acculturation. Given the importance of acculturation in mental health and quality of life [13,49], we hypothesized that (1) loneliness would be negatively predicted, whereas perceived social support would positively predict psychological well-being; and (2) acculturation would moderate the effects of loneliness or social support on psychological well-be.

## 2. Method

### 2.1. Participants

A final sample of 168 older Chinese immigrants (aged 65–89, *M* = 74.5, *SD* = 5.56, 106 women), with an average of 12 years living in Canada (144 in the Great Toronto Area, 12 in Montreal, and 12 in Edmonton), participated in this study from September to November of 2020. The recruitment flyer and/or the registration Google form were distributed through WeChat, the internet, or emails. Participants were then contacted to schedule a testing session either individually (*n* = 5) or in a group (*n* = 163). Participants received a $10 gift card (Walmart or Amazon) as a token compensation. Across the reading, writing, listening, and speaking domains, an average of 30% of the sample reported “no”, 39% reported “a bit”, and 20% reported “above-average” in English proficiency.

### 2.2. Measures

The survey was built in Qualtrics and includes the following measures in order: The demographic Questionnaire collects information such as age, gender, marital, education, employment status, and living arrangements as potential covariates. The Vancouver Index of Acculturation (VIA) [60] is a 20-item scale for enculturation (i.e., the practice of heritage culture; for example, “I enjoy social activities with Chinese people”) and acculturation (i.e., the practice of hosting culture; for example, “I enjoy social activities with typical North American people”), based on a 9-point Likert scale from 1 (strongly disagree) to 9 (strongly agree). Both sub-scales showed strong internal reliability (*a* = 0.91 for enculturation, *a* = 0.89 for acculturation), and they were used as moderators in the current study. The UCLA Loneliness Scale (Version 3) [61] is a 20-item scale for subjective feelings of loneliness and social isolation based on a 4-point Likert scale from 1 (never) to 4 (often). The total score ranges from 20 to 80, with a higher score indexing a higher level of loneliness. It showed decent internal reliability (*a* = 0.60) in the current study. The Multidimensional Scale of Perceived Social Support Scale (MSPSS) [62] is a 12-item scale assessing social support from significant others, family, and friends based on a 7-point Likert scale ranging from 1 (very strongly disagree) to 7 (very strongly agree). The total score ranges from 12 to 84, with a higher score indexing a higher level of perceived support. It showed high internal reliability (*a* = 0.91) in this study. Loneliness and Social support were used as predictors in this study. The World Health Organization-Five Well-Being Index (WHO-5) [63] has 5 items assessing emotional psychological well-being during the past two weeks based on a 6-point Likert Scale from 0 (at no time) to 5 (all of the time). The score (4 times the raw sum score) ranged from 0 (worst) to 100 (best), with a higher score meaning better well-being. The current sample showed high reliability (*a* = 0.88). The Satisfaction with Life Scale (SWLS) [64] is a 5-item scale for the global perception of life satisfaction based on a 7-point Likert scale ranging from 1 (strongly disagree) to 7 (strongly agree). The sum score (ranging 5–35) indexes life satisfaction, with a higher score meaning higher satisfaction. The current sample showed strong internal reliability (*a* = 0.85). WHO-5 and SWLS scores were used as dependent variables in this study.

### 2.3. Procedure

Informed consent was collected before the session starts. The data collection testing session took around 35 min, facilitated by research assistants through a Zoom or a phone meeting. For the individual testing session, the survey was largely completed by a research assistant on behalf of participants based on their spoken responses. At group testing Zoom sessions, the survey was largely completed by participants on their own through a distributed survey link through Zoom chat. Eight group testing sessions were conducted. Participants were randomly assigned a subject number emailed to them ahead of the scheduled session, and also reminded during the session if needed. They first received instructions on the testing procedure and were given opportunities to ask clarification questions. The survey link was then posted in the Zoom chat. Those who have questions or difficulty completing on their own were assigned to a private breakout room to get it completed, each with an individual research assistant.

### 2.4. Data Cleaning and Analysis

During the data cleaning, we replaced all the missing values (approximately 1% in total data points) by the average on each scale or subscale for each participant. Only those who completed at least 80% of the items of each scale were included in the final analysis on that scale [65]. We then checked the distribution normality of each outcome variable through skewness and kurtosis reported in the histograms and normal Q–Q plots. The Winserizing procedure was applied to replace all the outliers identified in the Stem-and Leaves Plots (2.5 SDs away from the group mean) with either the closest minimum or maximum value [66]. No violation of homogeneity and collinearity was detected based on the Levene’s test. All variables with a normal distribution were centered to reduce multi-collinearity [67] before being included in the regression analysis. The data analysis was conducted in IBM SPSS 24. We first performed univariate analysis of variance models (ANOVAs) on the two psychological well-being outcome scores stratified by sociodemographic variables (Table 1). These analyses aimed to describe the sample and identify potential sociodemographic predictors to serve as covariates in the subsequent regression models). As per convention [68], variables with a *p* ≤ 0.20 from the ANOVAs were identified as potential covariates in the subsequent corresponding multiple linear regression models (Table 2 and Table 3).

## 3. Results

### 3.1. Sample Demographic Characteristics and Group Differences in Outcome Variables

The sample characteristics and the group differences in outcome variables were displayed in Table 1. Based on the univariate ANOVA results, the variables with *p* ≤ 0.20 (i.e., gender, income, and religion for WHO-5 and education level for SWLS) were identified as potential sociodemographic covariates in the subsequent regression models.

### 3.2. Regression Analyses

Hierarchical regression models were conducted to identify significant predictors for the two psychological well-being outcomes: WHO-5 and SWLS, each with a 2-step model. In Step 1, loneliness and social support, together with sociodemographic covariates, were entered. In Step 2, acculturation and enculturation scores were added as moderators. 

#### 3.2.1. Regression on WHO-5 Score

This regression model examined the prediction of loneliness and perceived social support on WHO-5 score (i.e., emotional psychological well-being), controlling for potential demographic covariates identified in Table 1. In Step 1, loneliness and social support were entered along with the following covariates: income level, gender, and religion. The results showed a negative prediction of loneliness (*β* = −3.11, *p* = 0.002) and a positive prediction of social support (*β* = 1.04, *p* = 0.015) for the WHO-5 score. Income level negatively predicted (*β* = −1.08, *p* = 0.040), but neither gender (*β* = 0.35, *p* = 0.540) nor religion (*β* = 0.81, *p* = 0.160) predicted the WHO-5 score. In Step 2, the model was rendered non- significant (*F* = 0.81, *p* = 0.446). However, loneliness (*β* = −3.24, *p* = 0.001), social support (*β* = 0.91, *p* = 0.040), and income level (*β* = −1.14, *p* = 0.032) remained to be significant predictors. All the other variables did not show significant prediction (*p* ≥ 0.160).

#### 3.2.2. Regression on SWLS Scores

This regression model examined the prediction of loneliness and perceived social support for life satisfaction (i.e., cognitive component of psychological well-being), controlling for education level, a potential sociodemographic covariate identified in Table 1. In Step 1, loneliness and social support were entered, along with educational level. The results showed a negative prediction of loneliness (*β* = −0.52, *p* = 0.015) and a positive prediction of social support (*β* = 0.35, *p* < 0.000) for life satisfaction. Education level showed a significant prediction (*β* = 0.27, *p* = 0.048). With acculturation/enculturation added in Step 2, the model was rendered non-significant (*F* = 0.49, *p* = 0.620). Loneliness (*β* = −0.54, *p* = 0.013) and social support (*β* = 0.33, *p* = 0.000) remained to be significant predictors, but the prediction of education level became non-significant (*β* = 0.26, *p* = 0.040). Neither acculturation (*β* = 0.06, *p* = 0.420) nor enculturation (*β* = 0.04, *p* = 0.640) predicted life satisfaction.

## 4. Discussion

The current study confirmed that some critical psychosocial profiles such as loneliness and perceived social support serve as significant predictors for psychological well-being, as indexed by global emotional well-being (i.e., WHO-5) and cognitively perceived life satisfaction (i.e., SWLS) among Chinese immigrant older adults in Canada amidst COVID-19. Specifically, loneliness is a risk factor, whereas perceived social support serves as a protective buffer for psychological well-being.

Quarantine and social distancing practices not only restricted the spread of the virus but also generated a sense of social disconnection and loneliness [69]. Consistent with the previous studies [23,24,25,26], we found that loneliness negatively predicted both emotional psychological well-being and cognitively perceived life satisfaction. Individuals who experienced social isolation and loneliness were less likely to attend social activities, more likely to show depressive symptoms, and thus likely to be unsatisfied with their life [18,26]. This result highlights the importance of social engagement and social connection among older adults to reduce loneliness and thus promote psychological well-being and mental health during the pandemic.

Consistent with previous studies [15,24,29], the results identified perceived social support as a strong protective predictor for psychological well-being. Past research showed that people with positive psychological well-being can better cope with life challenges and, thus, tend to live a healthier and longer life [41,42]. This suggests that perceived social support plays an important role in the psychological well-being of Chinese seniors in Canada during the COVID-19 pandemic. This is probably because perceived social support is one important coping resource that helps older adults deal with adversities [56]. It has been shown that interventions enhancing social relationship quality can improve health and survival through stress and loneliness reduction [70]. Considering that a majority (i.e., 9/10) of older adults in Canada lived in a private household prior to COVID-19 [7], they probably largely rely on their family, friends, or even neighbors to take care of their groceries or essential needs during the pandemic. This might be more pronounced in older Chinese adults, considering their collectivism culture that promotes extended family structure, filial piety, and inter-dependency, as well as their language and cultural barriers in accessing external services or support beyond their families, social networks, or Chinese communities.

Our results further showed that those who perceived higher level of social support tended to be more satisfied with their life during the COVID-19 pandemic. Similarly, it has been found that social networks of friends and family contributed to older adults’ life satisfaction [14]. Life satisfaction is an overall assessment of one’s feelings, attitudes, and behaviors [15,71]. It has been shown that the life satisfaction of senior immigrants is positively correlated with overall health and a sense of belonging [20]. Taken together, higher perceived social support predicts better psychological well-being in both emotional and cognitive aspects.

Additionally, the current study also identified income level as a positive predictor for emotional psychological well-being. This result was consistent with the previous findings that unstable incomes have a great impact on mental health and well-being [72]. Lack of income may contribute to inappropriate healthcare utilization [73] and, thus, reduces their psychological well-being. The results were also consistent with a recent study that lower-income people experience greater isolation and a lower sense of belonging [69]. Moreover, our results also showed that lower education (< college) predicted higher life satisfaction. This was consistent with the previous finding of a negative relationship between education level and life satisfaction [74,75]. It should be noted that the results are mixed because other studies showed an opposite positive relationship [76,77]. It is probably that those older Chinese immigrants with lower than college education might have lower life expectations and be more likely to be dependent upon their family for social and financial support. Thus, their life might not be dramatically disrupted during the pandemic. This may buffer the detrimental psychological impacts of the pandemic.

However, against our hypotheses, the results did not find any moderation effect of acculturation or enculturation on the predictive effects of loneliness or social support on psychological well-being. This is possibly because older Chinese adults tend to live with their children and rely on their families for essential social and practical support, given their language and cultural barriers. Notably, over a quarter of the current sample has zero knowledge in English. Additionally, we did find that enculturation was significantly positively correlated with perceived social support (*r =* 0.22, *p* < 0.001). Specifically, those who were more identified with their heritage culture were more likely to perceive higher social support during the pandemic. Furthermore, it is also possible that the minimal variance in acculturation may have restricted its moderation effect.

This study has several limitations. First, it is impossible to draw causal-effect conclusions given the correlational study design. Second, the convenient sampling procedure does not guarantee the representativeness of our sample for the target population. Given the digital recruitment and online testing nature, the results might be biased toward those who are comfortable using digital devices. This would limit the generation of our results to the general older Chinese immigrant population. Thus, caution should be taken in making general conclusions. This might be considered by using multiple ways to avoid sampling biases in the future. Third, the self-report nature of the current study might result in interpretation biases. Fourth, the study focused only on older Chinese adults. It will be interesting to examine whether the reported results would be generalized to other minority aging populations.

Nevertheless, the current study makes novel contributions to the literature on aging. Specifically, we captured both the emotional and cognitive aspects of psychological well-being by using both the WHO-5 (as an index of emotional psychological well-being) and SWLS (as an index of cognitive psychological well-being). Furthermore, we assessed not only the risk factor (i.e., loneliness) but also the buffer factor (i.e., social support) for psychological well-being, and both factors were closely relevant to the context of the pandemic. The results suggest that the psychological well-being of older Chinese adults in Canada is negatively predicted by loneliness but positively predicted by perceived social support. The results highlight the psychosocial implications for the psychological well-being of the older immigrant population. The present study has important practical implications. First, considering that psychological well-being was related to low loneliness and higher perceived social support amidst COVID-19, the prevention and intervention programs for seniors may aim to enhance their social networks and decrease their loneliness. Immigrants during migration and settlement are facing a variety of challenges, such as unstable employment, language barriers, lack of social support, separation from families, and financial difficulties [34]. Seniors may have a greater need for social support to overcome their social isolation and promote their resilience under COVID-19. It is essential for their family members, relatives, or friends to be supportive to minimize their feeling of loneliness and increase their sense of social support. Additionally, it is crucial to develop language and culturally appropriate programs to improve life satisfaction and thus promote psychological well-being during the pandemic.

## 5. Conclusions

Taken together, the results identified loneliness as a risk factor, whereas perceived social support as a protective buffer factor for both emotional and cognitive psychological well-being among older Chinese immigrants during the pandemic. Furthermore, lower income is related to lower emotional psychological well-being, whereas a lower education level is related to higher cognitive psychological well-being (i.e., life satisfaction). Finally, acculturation does not moderate the effect of loneliness and social support on psychological well-being, probably because most older Chinese immigrants largely rely on family and friends for social and financial support due to language and cultural barriers. These findings provide an empirical foundation for government, policymakers, and stakeholders to best support minority senior groups with effective and culturally appropriate social support programs to mitigate feelings of loneliness, enhance social support, and thus promote the psychological well-being of older Chinese immigrants in Canada amidst the pandemic.

## Figures and Tables

**Table 1 ijerph-19-08612-t001:** Sample characteristics and their relationship with the two outcome variables: WHO-5 and SWLS (*n* = 168).

Variables		Sample Size	WHO-5			SWLS		
		*n* (%)	*M* (*SD*)	*F*	*p*	*M* (*SD*)	*F*	*p*
Age group	65–75	97 (57.1)	14.19 (3.77)	0.40	0.84	5.26 (0.78)	0.03	0.85
	≥76	63 (37.5)	14.07 (3.30)			5.28 (0.82)		
Gender	Male	53 (31.5)	14.54 (3.37)	2.85	0.06	5.25 (0.76)	0.74	0.48
	Female	106 (63.1)	13.68 (3.67)			5.26 (0.81)		
Marital status	Single	9 (5.4)	13.87 (4.36)	0.15	0.93	5.60 (0.83)	1.06	0.37
	Married	120 (71.4)	14.12 (3.56)			5.27 (0.80)		
	Divorced	7 (4.2)	13.60 (4.00)			4.91 (0.41)		
	Widowed	30 (17.9)	14.45 (3.56)			5.34 (0.78)		
Income	Low	82 (48.8)	13.48 (3.54)	5.08	0.03	5.20 (0.86)	1.55	0.22
	Middle	86 (51.2)	14.71 (3.55)			5.35 (0.71)		
Place of origin	Mainland	159 (94.6)	14.08 (3.59)	0.66	0.62	5.28 (0.79)	0.41	0.80
	Hongkong	6 (3.1)	13.20 (4.52)			5.30 (0.86)		
	Taiwan	1 (1.0)	13.60 (0.00)			4.60 (0.00)		
Years living in Canada	<6 years	31 (19)	14.01 (3.67)	0.03	0.87	5.26 0(.89)	0.00	0.97
	≥ 6 years	133 (78.6)	14.13 (3.63)			5.27 (0.77)		
Education level	≥college	116 (69)	14.30 (3.57)	0.97	0.33	5.20 (0.81)	3.51	0.06
	<college	52 (31)	13.71 (3.63)			5.45 (0.72)		
Citizenship	Citizen	39 (23.2)	14.42 (3.32)	0.17	0.84	5.33 (0.78)	0.22	0.81
	Permanent	119 (70.8)	14.08 (3.67)			5.27 (0.79)		
Employment	Retired	151 (89.9)	14.07 (3.65)	0.24	0.87	5.29 (0.79)	0.57	0.64
	Full-time	4 (2.4)	14.60 (4.68)			4.95 (0.84)		
	Self-employed	1 (0.6)	12.00 (0.00)			6.00 (0.00)		
Religion	No	114 (67.9)	14.39 (3.68)	1.75	0.19	5.30 (0.79)	0.22	0.64
	Yes	52 (30.1)	13.60 (3.37)			5.23 (0.80)		

Note: M = mean, SD = Standard Deviation. Variables with *p* ≤ 0.20 would be entered in the regression models as covariates (see Table 2 and Table 3).

**Table 2 ijerph-19-08612-t002:** Hierarchical regression on emotional psychological well-being (i.e., WHO-5).

Model	Predictors	*β*	95% *CI* (Lower, Upper)	*R^2^*	*F*
Step 1 Model				0.14	5.06 ***
	Loneliness	−3.11 **	−5.04, −1.18		
	Social Support	1.04 *	0.21, 1.87		
	Income				
	Middle (reference)				
	Low	−1.08 *	−2.11, −0.04		
	Gender				
	Female (reference)				
	Male	0.35	−0.78, 1.47		
	Religion				
	Yes (reference)				
	No	0.81	−0.32, 1.94		
Step 2 Model				0.14	0.81
	Loneliness	−3.24 **	−5.19, −1.29		
	Social Support	0.91 *	0.04, 1.77		
	Income				
	Middle (reference)				
	Low	−1.14 *	−2.19, −0.10		
	Gender				
	Female (reference)				
	Male	0.42	−0.72, 1.55		
	Religion				
	Yes (reference)				
	No	0.81	−0.32, 1.94		
	Acculturation	0.31	−0.33, 0.94		
	Enculturation	0.31	−0.52, 1.13		

CI = confidence interval; * *p* < 0.05, ** *p* < 0.01, and *** *p* < 0.001.

**Table 3 ijerph-19-08612-t003:** Hierarchical regression on life satisfaction (i.e., SWLS).

Model	Predictors	*β*	95% *CI* (Lower, Upper)	*R^2^*	*F*
Step 1 Model				0.13	8.31 ***
	Loneliness	−0.52 *	−0.94, −0.10		
	Social Support	0.35 ***	0.17, 0.53		
	Education				
	≥college (reference)				
	<college	0.27 **	0.02, 0.51		
Step 2 Model				0.14	0.49
	Loneliness	−0.54 *	−0.97, −0.12		
	Social Support	0.33 ***	0.14, 0.52		
	Education				
	≥college (reference)				
	<college	0.26 *	0.01, 0.51		
	Acculturation	0.06	−0.08, 0.20		
	Enculturation	0.04	−0.14, 0.22		

CI = confidence interval; * *p* < 0.05, ** *p* < 0.01, and *** *p* < 0.001.

## Data Availability

The data files can be retrieved from https://doi.org/10.17605/OSF.IO/U2V8W (accessed on 8 July 2022).

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
