# Peer review of "The Psychological Well-Being of Older Chinese Immigrants in Canada amidst COVID-19: The Role of Loneliness, Social Support, and Acculturation"

_ijerph, 2022, doi:10.3390/ijerph19148612_

Round 1
Reviewer 1 Report
Abstract
How can “life satisfaction effect the psychological well-being “
Are you measuring the links between two dependent variables, both are relevant to wellbeing? Please define again your aims and hypothesis
Introduction
The introduction is lacking a strong and unifying theoretical framework that will connect all the paragraphs starting from line 47
Theories of health or well being, resources using Hobfull concepts? You need to reorganize the background according to the theoretical concepts/
Life satisfaction line 84 -- Please reread the original conceptualization of Diener and refer to it. Explain the links between these positive properties all part of the domain of positive psychology.
The issue of immigrants do not appear in the title- repharse and relate to it also by viewing the grand picture of resources.
All in allת you have two pathways – the first is mediated by stress the other by resources. Both are related to wellbeing.
Method abstract
In a cross sectional study design you do not predict but rather examine links and association. The session took around 35 minutes, facilitated by – 133 which session? Data collection?
Separate procedure from measurement . Expand on the scales.
It is written badly with no separation between the type of variables- independent , dependent? Possible mediators
Data analysis need to be rewritten according to the study hypothesis. It will be easier to follow such a presentation also in the result section.
Results
Add Tukey or another test to the ANOVA . explain.
Discussion
Need to be totally rewritten according to the theoretical contribution. It is currently impossible to understand what is important and what is negligible.
You need the expand on your limitation and future studies. Add references from studies all over the world that intervene with elderly people via zoon in the COVID 19 pandemic . Add studies that examine loneliness and depression , coping and other resources that improves via zoom intervention. What was found there and what are your suggestions.
Author Response
July 7, 2022
RE: ijerph-1775576 revision
Dear Editor and Reviewers,
Thank you for your thoughtful review of the manuscript ijerph-1775576. We revised the manuscript to address all the comments raised by the two reviewers. All of the changes are highlighted in red font in the revised manuscript. Our responses to the comments are outlined in our responses to each reviewer.
We believe the revisions effectively addressed all the comments raised in the reviewers. We look forward to your further evaluation and hope the manuscript will be accepted for publication. Please let us know if you have any further questions or comments.
We thank you for your time in reviewing our manuscript.
Kind regards,
Lixia Yang
Professor
Department of Psychology
Toronto Metropolitan University
350 Victoria St., JOR918
Toronto, Ontario, M5B 2K3
Tel: 416-979-5000 ext: 6522
Fax: 416-979-5273
e-mail: lixiay@ryerson.ca
Web: http://psychlabs.ryerson.ca/cal/
Response to the Comments Raised by Reviewer 1
Manuscript ID: ijerph-1775576
Point 1. Abstract. How can “life satisfaction effect the psychological well-being”, Are you measuring the links between two dependent variables, both are relevant to wellbeing? Please define again your aims and hypothesis.
Response: Following this comment, we introduced life satisfaction as the cognitive aspect of psychological well-being (p. 2) and used it as a dependent outcome variable in the revision. The research questions and hypotheses were rephrased accordingly (p. 3). We also re-analyzed the data to test the prediction of loneliness and social support for both WHO-5 and SWLS, indexing emotional and cognitive psychological well-being respectively (p. 5-7). The introduction was reframed accordingly.
Point 2. The introduction is lacking a strong and unifying theoretical framework that will connect all the paragraphs starting from line 47. Theories of health or well-being, resources using Hobfull concepts? You need to reorganize the background according to the theoretical concepts.
Response: Following this comment, we re-structured the introduction to enhance the coherence and transition. The individual concepts were introduced sequentially in a consistent order (same as in the other sections of the manuscript). For example, general emotional psychological wellbeing followed by life satisfaction).
Point 3. Life satisfaction line 84 -- Please reread the original conceptualization of Diener and refer to it. Explain the links between these positive properties all part of the domain of positive psychology.
Response: We clarified the definition of life satisfaction based on Diener’s theory, specifically as a cognitive component of subjective well-being (p. 2). We also described the relationship of life satisfaction with other positive attributes in the introduction (p. 2). In the revision, life satisfaction was used as a dependent variable instead of a moderator to index the cognitive aspect of psychological well-being (p. 2). We also cited Diener as reference 12 (p. 2). References were updated accordingly.
Point 4. The issue of immigrants do not appear in the title- rephrase and relate to it also by viewing the grand picture of resources..
Response: We added “immigrant” to the title and the related sections in the manuscript as per this comment (e.g., title, p. 1 abstract and p. 3 questions).
Point 5. All in allת you have two pathways – the first is mediated by stress the other by resources. Both are related to wellbeing.
Response: That is right, we highlighted that loneliness and social support as a risk and protective predictor respectively for psychological well-being in the revision.
Point 6. In a cross sectional study design you do not predict but rather examine links and association. The session took around 35 minutes, facilitated by – 133 which session? Data collection?
Response: We assume that regression analysis allows us to draw conclusions on predictive relationship. But we did point out this limitation of the cross-sectional design (p. 8). We also clarified the session as “testing session” (p. 3).
Point 7. Separate procedure from measurement . Expand on the scales. It is written badly with no separation between the type of variables- independent , dependent? Possible mediators
Response: Following the comments, we separated the procedure section (section 2.3) from the measure section (section 2.2) (p. 3-4). We also clearly classified and identified the measures into independent, dependent (outcome), and moderator variables (p. 3-4).
Point 8. Data analysis need to be rewritten according to the study hypothesis. It will be easier to follow such a presentation also in the result section.
Response: We streamlined the data analysis and results section to be consistent with the research questions and hypotheses in the introduction. The information was presented consistently in the same order across sections.
Point 9. Results. Add Tukey or another test to the ANOVA . explain.
Response: We streamline the data analysis and start the session with a sample group differences in each of the two outcome variables (WHO-5 and SWLS) with a set of univariate ANOVAs (p. 4-5).
Point 10. Discussion needs to be totally rewritten according to the theoretical contribution. It is currently impossible to understand what is important and what is negligible.
Response: We restructured the discussion according to the new data analysis and highlighted the theoretical contributions in the final paragraph (p. 8-9).
Point 11. You need the expand on your limitation and future studies. Add references from studies all over the world that intervene with elderly people via zoon in the COVID 19 pandemic . Add studies that examine loneliness and depression , coping and other resources that improves via zoom intervention. What was found there and what are your suggestions.
Response: We did streamline the discussion to clearly identify critical limitations and identified some future directions (p. 8). References were updated to capture the research across the globe.
Reviewer 2 Report
From my review of the manuscript “The Psychological Well-being of Chinese Older Adults in Canada amidst COVID-19: The Role of Loneliness, Social Support, and Life Satisfaction” I can say that it covers a very interesting and relevant topic; and has several implications. The paper is well written and meets the required scientific rigor. There are only minor points that I would like to highlight as possible suggestions for improvement.
In the method section:
- It would be important to say how many elements were in the group sessions.
- Correct the “strong internal reliability” for the VIA (a = .60 for enculturation, a = .60 for acculturation) (line 141).
- Correct the name of the scale to “The World Health Organization-Five Wellbeing Index” (line 152).
In the results section:
- I suggest not using bold to highlight statistically significant results (as present on table 1)
In the discussion section:
- The purpose of evaluate if acculturation would moderate the effects of loneliness or social support on psychological well-being is not properly considered. The discussion focused is only focused on the effect of life satisfaction.
- The relationship of life satisfaction and psychological well-being was already somehow expected, even by the similarity of both measures. It would be important to discuss this issue.
- Since the procedures of recruitment were digital (WeChat, the internet, or emails), will not be this sample too biased? And all those Chinese older people who do not use internet? Some considerations about this effect must be presented. Even because they must have different conditions of acculturation.
Author Response
July 7, 2022
RE: ijerph-1775576 revision
Dear Editor and Reviewer,
Thank you for your thoughtful review of the manuscript ijerph-1775576. We revised the manuscript to address all the comments raised by the two reviewers. All of the changes are highlighted in red font in the revised manuscript. Our responses to the comments are outlined in our responses to each reviewer.
We believe the revisions effectively addressed all the comments raised in the reviewers. We look forward to your further evaluation and hope the manuscript will be accepted for publication. Please let us know if you have any further questions or comments.
We thank you for your time in reviewing our manuscript.
Kind regards,
Lixia Yang
Professor
Department of Psychology
Toronto Metropolitan University
350 Victoria St., JOR918
Toronto, Ontario, M5B 2K3
Tel: 416-979-5000 ext: 6522
Fax: 416-979-5273
e-mail: lixiay@ryerson.ca
Web: http://psychlabs.ryerson.ca/cal/
Response to the Comments Raised by Reviewer 2
Manuscript ID: ijerph-1775576
Point 1. In the method section, it would be important to say how many elements were in the group sessions.
Response: Following this comment, we clearly listed all the measures included in the survey and classified them into covariates, moderators, predictors and dependent variables (p. 3-4).
Point 2. Correct the “strong internal reliability” for the VIA (a = .60 for enculturation, a = .60 for acculturation) (line 141).
Response: Corrected as “Both sub-scales showed strong internal reliability (a = .91 for enculturation, a = .89 for acculturation) (p. 3).
Point 3. Correct the name of the scale to “The World Health Organization-Five Wellbeing Index” (line 152).
Response: Corrected (p. 4).
Point 4. In the results section, I suggest not using bold to highlight statistically significant results (as present on table ).
Response: Thank you for the suggestion. We reframed Table 1 to present group difference ANOVA results and used a cut-off of p ≤ .20 to identify potential predictors for the two psychological well-being index scores. We removed the bold font from the p values.
Point 5. In the discussion section, the purpose of evaluate if acculturation would moderate the effects of loneliness or social support on psychological well-being is not properly considered. The discussion focused is only focused on the effect of life satisfaction.
Response: Life satisfaction is now used as a dependent outcome variable in the revision. We removed the related discussion from the discussion section. We also added more discussion on the lack of the effect of acculturation (p. 8).
Point 6. The relationship of life satisfaction and psychological well-being was already somehow expected, even by the similarity of both measures. It would be important to discuss this issue.
Response: After some deliberation and considering the comments from reviewer 1 as well, we changed life satisfaction to be a dependent outcome variable to index the cognitive component of psychological well-being in the revision. It makes a more coherent story and is consistent with the theoretical consideration of life satisfaction in literature. We reframed the introduction and discussion accordingly.
Point 7. Since the procedures of recruitment were digital (WeChat, the internet, or emails), will not be this sample too biased? And all those Chinese older people who do not use internet? Some considerations about this effect must be presented. Even because they must have different conditions of acculturation.
Response: We discussed this potential bias in the discussion section (p. 8).
Round 2
Reviewer 1 Report
This version is much better